# LINK WEIGHT PREDICTION WITH NODE EMBEDDINGS

## ABSTRACT

Application of deep learning has been successful in various domains such as image recognition, speech recognition and natural language processing. However, the research on its application in graph mining is still in an early stage. Here we present the first generic deep learning approach to the graph link weight prediction problem based on node embeddings. We evaluate this approach with three different node embedding techniques experimentally and compare its performance with two state-of-the-art non deep learning baseline approaches. Our experiment results suggest that this deep learning approach outperforms the baselines by up to 70% depending on the dataset and embedding technique applied. This approach shows that deep learning can be successfully applied to link weight prediction to improve prediction accuracy.

## 1 INTRODUCTION

Deep learning has outperformed other machine learning techniques in various application domains, e.g., speech recognition, image recognition, and natural language processing (Simonyan & Zisserman, 2014) (Yao et al., 2013). Deep learning can not only achieve higher prediction accuracy than other machine learning techniques, but also require less domain knowledge and feature engineering. Graph mining is a new and active domain for deep learning (Grover & Leskovec, 2016). An important problem in graph mining is link weight prediction. This problem has a state of the art non deep learning solution called Stochastic Block Model (SBM) (Aicher et al., 2014).

The contribution of this paper is to introduce the first generic deep learning approach to the link weight prediction problem based on node embeddings. This approach requires an effective node embedding technique to produce node embeddings. We demonstrate how this approach works with various embedding techniques including locally linear embedding (Roweis & Saul, 2000), skip-gram model (Grover & Leskovec, 2016) and Model R (Hou & Holder, 2017). We also show that this generic deep learning approach outperforms the SBM approach.

## 2 PROBLEM

We consider the problem of link weight prediction in a weighted directed graph. We first show an example of the problem, and then give the problem definition. An undirected graph can be reduced to a directed graph by converting each weighted undirected link to two directed links with the same weight and opposite directions. So the prediction for a weighted undirected graph is a special case of the problem we consider.

### 2.1 PROBLEM EXAMPLE

Let us look at an example of link weight prediction: message volume prediction in a social network. In this example, there are three users in a social network: A, B and C. Each user can send any amount of text messages to every other user. We know the number of messages transmitted between A and C, B and C, but not A and B. We want to predict the number of messages transmitted between A and B. The ability to predict these interactions potentially allows us to recommend new connections to users: if A is predicted/expected to send a large number of messages to B by some model, and A is not connected to B yet, we can recommend B as a new connection to A.

## 2.2 PROBLEM DEFINITION

The definition of the link weight prediction problem in a weighted directed graph is: given a weighted directed graph with the node set V and link subset E, build a model w = f(x, y) where x and y are nodes and w is the weight of link (x, y) that can predict the weight of any link For every possible link (1 out of $n^2$, where n is the number of nodes), if we know its weight, we know it exists; if we do not know its weight, we do not know if it exists. This is a very practical point when we handle streaming graphs: for any possible link, we either know it exists and know its weight (if it has been streamed in), or we do not know if the link will ever exist, nor know its weight.

## 3 EXISTING APPROACHES

In our literature study on previous research in the link weight prediction problem, we have found some existing approaches. In this section, we review these existing approaches.

### 3.1 SBM (STOCHASTIC BLOCK MODEL)

The SBM uses link existence information (Holland et al., 1983). The main idea is to partition nodes into L groups and connect groups with bundles. Each group consists of nodes which are topologically similar in the original graph, and groups are connected by bundles to represent the original graph. The SBM has the following parameters:

- z: the group vector, where $z_i \in \{1...L\}$ is the group label of node i
- $\theta$: the bundle existence probability matrix, where $\theta_{z_i z_j}$ is the existence probability of bundle $(z_i, z_j)$

Given a graph with adjacency matrix A, the existence of link (i, j) $A_{ij}$ is a binary random variable following the Bernoulli distribution:

$$A_{ij} \sim B(1, \theta_{z_i z_j})$$

The SBM fits parameters z and $\theta$ to maximize the probability of observation A:

$$P(A|z, \theta) = \prod_{ij} \theta_{z_i z_j}^{A_{ij}} (1 - \theta_{z_i z_j})^{1 - A_{ij}}$$

We rewrite the probability of observation A as a log likelihood:

$$\log(P(A|z, \theta)) = \sum_{ij} (A_{ij} \log(\frac{\theta_{z_i z_j}}{1 - \theta_{z_i z_j}}) + \log(1 - \theta_{z_i z_j}))$$

### 3.2 PWSBM (PURE WEIGHTED STOCHASTIC BLOCK MODEL)

The pWSBM is derived from SBM but it uses link weight information (Aicher et al., 2014). So it differs from SBM in a few ways described below. Adjacency matrix A becomes the link weight matrix where the weight of link (i, j) $A_{ij}$ is a real random variable following the normal distribution:

$$A_{ij} \sim N(\mu_{z_i z_j}, \sigma_{z_i z_j}^2)$$

$\theta_{z_i z_j}$ becomes the weight distribution parameter of bundle $(z_i, z_j)$:

$$\theta_{z_i z_j} = (\mu_{z_i z_j}, \sigma_{z_i z_j}^2)$$

The pWSBM fits parameters z and $\theta$ to maximize the log likelihood of observation A:

$$\log(P(A|z, \theta)) = \sum_{ij} (A_{ij} \frac{\mu_{z_i z_j}}{\sigma_{z_i z_j}^2} - A_{ij}^2 \frac{1}{2\sigma_{z_i z_j}^2} - \frac{\mu_{z_i z_j}^2}{\sigma_{z_i z_j}^2})$$

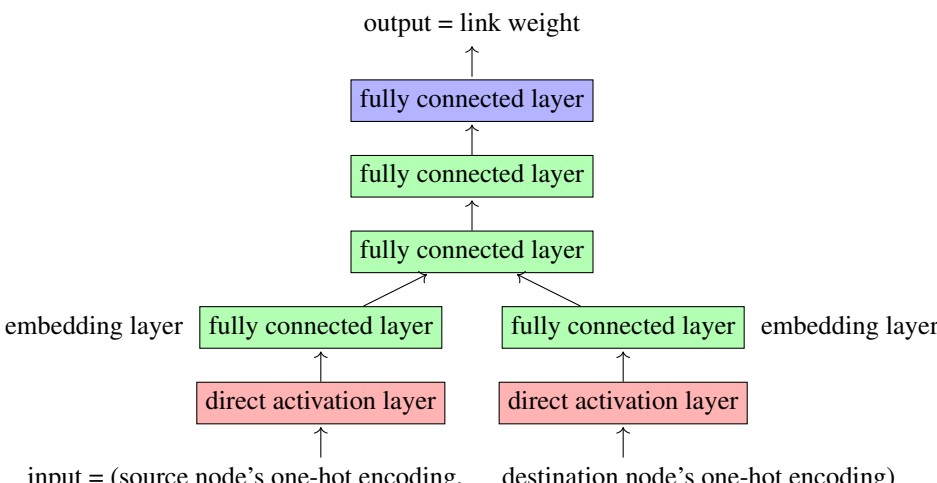

Figure 1: A simplified version of Model R with 1 input layer (red), 3 hidden layers (green), and 1 output layer (blue). The embedding layer and the input layer each has two channels: one channel for the source node and one channel for the destination node. Only layers and their connections are shown, while the units in each layer and their connections are not shown.

### 3.3 MODEL R

This model is a fully connected neural network (Hou & Holder, 2017). It uses an end-to-end supervised learning approach to predict link weights, as shown in Figure 1. The model contains the following layers:

- An input layer directly activated by the one-hot encodings of a (source node, destination node) pair.
- A hidden embedding layer of linear units. This layer maps each node from its one-hot encoding to the corresponding node vector.
- Multiple fully connected hidden layers of rectified linear units ($f(x) = \max(0, x)$, only two layers are shown in the figure).
- An output layer with a linear regression unit($f(x) = kx$).

## 4 OBSERVATIONS AND MOTIVATION

We are interested in decoupling two learning processes (node embedding learning and link weight prediction learning) in Model R to generalize it and design a generic deep learning approach. In other words, it mixes two learning processes into one single learning process. On the other hand, there are several other general purpose node embedding techniques such as Node2vec (Grover & Leskovec, 2016). So we wonder if the node embeddings produced by those general purpose node embedding techniques are better than node embeddings produced by Model R. If we can decouple the two learning processes in Model R, we can substitute the node embedding with any other node embedding technique. This plug-and-play idea, if it works, is very appealing because every time a new node embedding technique comes out, we can easily use the embeddings it produces in link weight prediction and see if it can improve the prediction accuracy.

## 5 APPROACH

We want to create a generic link weight prediction approach that uses the node embeddings produced by any node embedding technique. Given a weighted graph, from its adjacency list we can produce a dataset where each example is a (source node, destination node, link weight) triplet. For example, given a social network where nodes are users and link weights are numbers of messages users send

to other users, we have its adjacency list dataset as shown in Table 1, on which we will train the model.

Table 1: The adjacency list dataset for a social network.

| Input = (source, destination) | Output = weight |
|:---:|:---:|
| ... | ... |
| (Mary, John) | 8645 |
| (John, Mary) | 9346 |
| (John, Alice) | 2357 |
| (John, Bob) | 9753 |
| (Alic, Bob) | 1238 |
| ... | ... |

## 5.1 EMBEDDING TECHNIQUES

As deep learning techniques become more powerful and standardized, a key process of a domain-specific deep learning application is mapping entities to vectors in an embedding space, because a neural net needs vectors as inputs. These vectors are called embeddings and this process is called embedding. Embeddings are ubiquitous in deep learning, appearing in natural language processing (embeddings for words), graph mining (embeddings for nodes) and other applications. Embedding techniques learn these vectors from relations between words and nodes. A common goal of these techniques is to ensure that similar entities are close to each other in the embedding space.

### 5.1.1 WORD2VEC: WORD EMBEDDING WITH SKIP-GRAM MODEL

Word2vec is an unsupervised embedding technique commonly used in natural language processing (Mikolov et al., 2013). It maps every word in a vocabulary to a vector without any domain knowledge. A small skip-gram neural network model is shown in Figure 2. A natural language corpus has

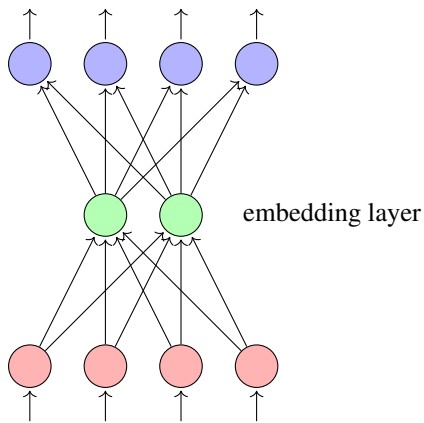

Figure 2: A small skip-gram neural network model with 1 input layer (red), 1 hidden layer (green), and 1 output layer (blue), vocabulary size 4 and embedding size 2. The hidden layer uses linear units. The output layer uses softmax units.

many sentences, therefore, from these sentences we can produce a dataset where each example is a (word, context-word) pair. For example, given the sentence "the quick brown fox jumps over the lazy dog", a context radius of 2, and the word "fox", we have the context of fox {quick, brown, jumps, over}. During each training step, one training example - a (word, context-word) pair - is used. The input layer is activated by the one-hot encoding of the given word. The output layer activation is the

context-word probability distribution. The embeddings for all words are technically the weights in the embedding layer. The final outcome is that similar words have similar context-word probability distributions and similar embeddings. Acquiring these embeddings is often the first step in many natural language processing tasks such as sentiment analysis (Socher et al., 2013) and information retrieval (Shen et al., 2014).

### 5.1.2 NODE2VEC: NODE EMBEDDING WITH SKIP-GRAM MODEL

Node2vec is a node embedding technique similar to word embedding (Grover & Leskovec, 2016). This technique reduces the node embedding problem to the word embedding problem and then applies the word embedding technique. It is a generalization of an earlier node embedding technique DeepWalk (Perozzi et al., 2014). More specifically, DeepWalk is a special case of Node2vec where the inbound parameter p and outbound parameter q are both set to 1. A graph has many walks, which can produce a dataset where each example is a (node, context-node) pair. For example, given a walk in a social network of users {John, Mary, James, Alice, Bob}, we have the context of James {John, Mary, Alice, Bob}. By reducing walks (sequences of nodes) to natural language sentences (sequences of words) and nodes to words, this technique reduces the node embedding problem to the word embedding problem. The final outcome is similar nodes have similar embeddings.

### 5.1.3 MODEL R: NODE EMBEDDING SUPERVISED BY LINK WEIGHTS

Model R not only learns to predict link weight, but also has its own node embedding technique, as we have shown in the previous section. In fact, Model R produces its own node embeddings. Model R based node embedding technique is different from the skip-gram based Node2vec technique. One advantage of the Model R based node embedding technique is that it takes advantage of the highly organized, regular and repeated structure in the relational dataset representing a graph, i.e., a source node connects to a destination node through one and only one weighted link. The skip-gram model does not exploit this structure in natural language processing because this structure does not exist.

### 5.1.4 LLE (LOCALLY LINEAR EMBEDDING): NONLINEAR DIMENSIONALITY REDUCTION

LLE is a member of the family of manifold learning approaches (Roweis & Saul, 2000). This technique is not specifically designed for any embedding task, but for dimensionality reduction. This technique comprises two steps. The first step is linear approximation of data points in the original high dimensional space, an optimization process where the cost function is

$$cost(W) = \sum_i |X_i - \sum_j W_{ij} X_j|^2$$

where each data point (high dimensional vector) $X_i$ is linearly approximated by its k nearest neighbors $X_j$'s, and the weight matrix W is the linear approximation parameter to be optimized. The weight matrix $W$ is determined by the intrinsic geometric properties of the neighborhoods instead of the coordinate system, therefore it obeys an important symmetry: $W$ is invariant to rotations, scalings, and translations in the coordinate system. The second step is reconstruction of data points in a new low dimensional space, an optimization process where the cost function is

$$cost(Y) = \sum_i |Y_i - \sum_j W_{ij} Y_j|^2$$

where each data point (low dimensional vector) $Y_i$ is linearly approximated by its k nearest neighbors $Y_j$'s with the same weight matrix W produced in the previous linear approximation step. Now the new coordinates Y are the parameters to be optimized. To apply this method, we use each row in the adjacency matrix of the graph as a data point in the high dimensional space, and then use LLE to map all rows to node embeddings in the low dimensional space.

### 5.2 MODEL S: LINK WEIGHT PREDICTION WITH NODE EMBEDDINGS

We introduce model S, a neural net model that predicts link weight using a pair of node embeddings (produced by any node embedding technique) as its input. Model S is a generalization of Model R. More specifically, Model R is a special case of Model S when the pair of node embeddings is produced by itself.

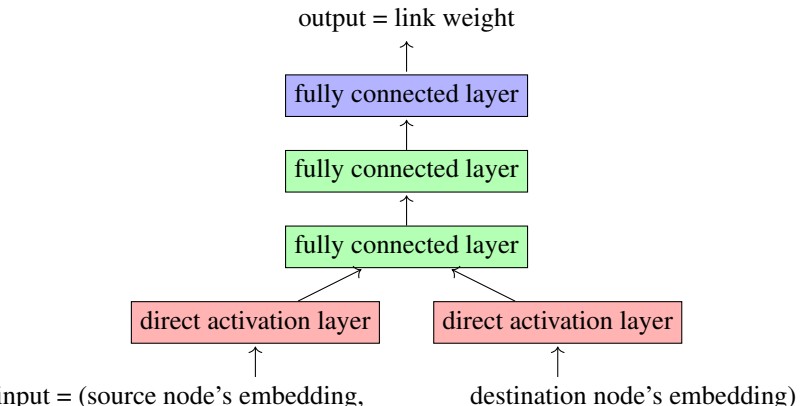

Figure 3: Model S with 1 input layer (red), 2 hidden layers (green), and 1 output layer (blue). The input layer has two channels: one channel for the source node and one channel for the destination node. Only layers and their connections are shown, while the units in each layer and their connections are not shown.

The approach uses the well-known deep learning techniques of back propagation (Rumelhart et al., 1988) and stochastic gradient descent (LeCun et al., 2012) in model training. Empirically, we use multiple hidden layers where each layer contains $d = log_2(n)$ hidden units where d (as in dimension) is the number of units in each hidden layer and n is the dataset size. We will vary the number of hidden layers to assess impact on performance. This approach updates the model at each training step so Model S is an online learning method as long as the node embedding technique it depends on is also an online learning method (every embedding method we discussed in this section except LLE is an online learning method). Therefore, by using an online embedding method, Model S can nicely handle streaming graphs.

## 6 EXPERIMENTS

We evaluate Model S experimentally with SBM with pWSBM as baselines, and compare their prediction errors on several datasets. We use three different techniques to produce node embeddings: node2vec, LLE and Model R. The results show that Model S outperforms the baselines with all three embedding techniques on three out of four datasets. Model S with Model R embedding technique performs the best on four out of four datasets.

### 6.1 DATASETS

The experiments use the following datasets:

- Airport (Colizza et al., 2007). Nodes represent the 500 busiest airports in the United States, and each of the 5960 edges is weighted by the number of passengers traveling from one airport to another.

- Authors (Newman, 2001). Nodes represent 16726 authors and each of the 47594 edges is weighted by the number of papers posted by two authors to Condensed Matter section of arXiv E-Print Archive between 1995 and 1999.

- Collaboration (Pan et al., 2012). Nodes represent 226 nations on Earth, and each of the 20616 edges is weighted by the number of academic papers whose author lists include that pair of nations.

- Congress (Porter et al., 2005). Nodes represent the 163 committees in the 102nd United States Congress, and each of the 26569 edges is weighted by the number of shared members.

- Facebook (Opsahl, 2013). Nodes represent 899 users of a Facebook-like forum, and each of the 71380 edges is weighted by the number of messages posted by two users on same topics.
- Forum (Opsahl & Panzarasa, 2009). Nodes represent 1899 users of a student social network at UC Irvine, and each of the 20291 edges is weighted by the number of messages sent between users.

We split each dataset into training set (80%) and testing set (20%). We normalize the link weights in each dataset to range [-1, 1] after applying a logarithm function. As mentioned in previous section, Model S is suitable as a online learning method for steaming graphs so it scales nicely with dataset size. More specifically, the time complexity of the training stage is n(E), where E is the number of links in the graph.

## 6.2 EVALUATION PROCESS

We do the same experiment for each dataset. We use mean squared error as the prediction accuracy metric. Each experiment consists of 25 independent trials. For each trial we train the model using the training set and evaluate the model on the testing set. For each experiment, we report the mean of the errors from 25 trials. The program is open-source under MIT license hosted on Github so that everyone can use it without any restriction [1].

## 6.3 MODEL PREDICTION ACCURACY

We evaluate the performance of Model S and compare it with a previously published performance evaluation of two baselines SBM and pWSBM on 4 datasets (Airport, Collaboration, Congress and Forum) (Aicher et al., 2014). We were not able to acquire an implementation for SBM or pWSBM so we did not do any performance evaluation on other datasets. The Model S's prediction accuracy is better than baselines on 3 out of 4 datasets. Among all Model S variants, the one with Model R embedding technique performs the best, as shown in Figure 4. One obvious reason Model S can

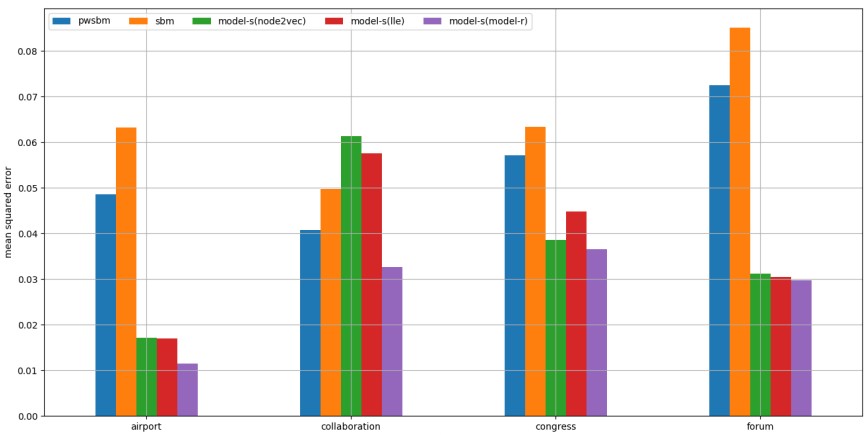

Figure 4: Model S performs better when using 3 different embedding techniques (LLE, Model R and node2vec) on 3 out of 4 datasets (Airport, Collaboration, Congress and Forum).

outperform baselines is higher level of flexibility. SBM based approaches assume nodes within the same group are the same in terms of link weight distributions and the weight of every link follows a normal distribution. Model S does not rely on these assumptions as it gives each node a unique description - the node embedding and it takes advantage of high flexibility of layers of non-linear neural network units. A possible reason why the Model S with Model R embedding outperforms

---

[1] https://github.com/xxxxxx

other two variants with node2vec and LLE embedding is Model R embedding is specifically designed for link weight prediction task.

## 6.4 MODEL ROBUSTNESS

In our experiments, Model S is robust against model parameter changes. Adding or removing a hidden layer or several units in a hidden layer usually does not change the prediction accuracy dramatically. For example, when we set the number of hidden units in each hidden layer to the same constant value 20 and change the number of hidden layers, the prediction error has little fluctuation (less than 10%) over a wide range of values for the number of hidden layers, as shown in Figure 5. We have similar results on other four datasets and with other parameters like number of units in each hidden layer.

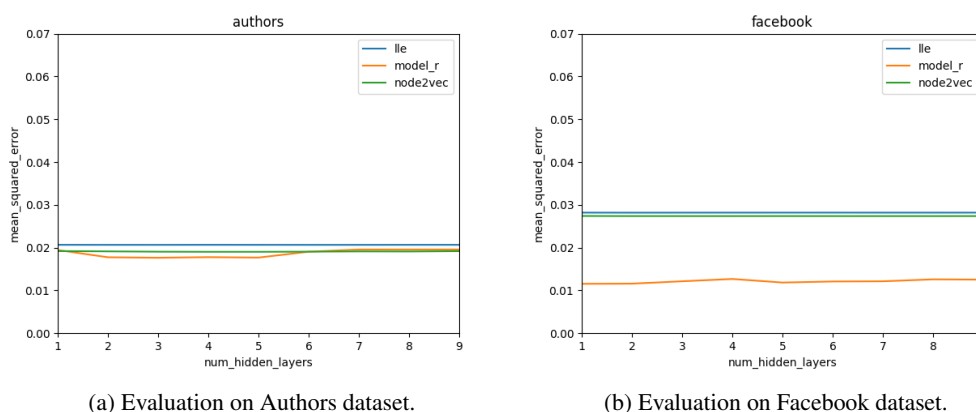

(a) Evaluation on Authors dataset.          (b) Evaluation on Facebook dataset.

Figure 5: Model S is robust against changes of the model parameter number of hidden layers when using 3 different embedding techniques (LLE, Model R and node2vec) on 2 datasets (Authors and Facebook).

## 7 CONCLUSION

Model S is the first generic deep learning approach to the link weight prediction problem based on node embeddings. It requires an effective node embedding technique to supply it node embeddings. It works with various embedding techniques including LLE, node2vec and the Model R. In most cases, its prediction accuracy is better than the state-of-the-art non deep learning approaches SBM and pWSBM. The Model R embedding performs the best among the different embeddings, which indicates that embeddings learned in the context of the task they will be used for can improve performance over generic embedding methods.

## 8 FUTURE WORK

A potential direction for this work is to identify metrics for evaluating the learned node embeddings. A possible metric is the distances of the embeddings of similar nodes. Ultimately, good embeddings should produce good prediction accuracy, because eventually some other machine learning system should use these embeddings to do valuable predictions. Therefore, good metrics should have positive correlation with prediction accuracy of the later stages of the deep learning system whose inputs are the embeddings.

### ACKNOWLEDGMENTS

This material is based upon work supported by the National Science Foundation under Grant No. xxxxxx.

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
