# OpenReview forum: "Link Weight Prediction with Node Embeddings"
_ICLR.cc/2018/Conference — Reject_

### Official Review · AnonReviewer3 · 2017-11-26
**This paper aims at tackling the weight prediction problem for potential links in graphs, and proposes a deep-learning approach (called model S) to solve this particular problem. The proposed model is formulated as a multi-layered neural network, which consider two node embeddings of the vertices on both side of edges.**

**Rating:** 3
**Confidence:** 4

**Review:**

Although this paper aims at an interesting and important task, the reviewer does not feel it is ready to be published.
Below are some detailed comments:

Pros
- Numerous public datasets are used for the experiments
- Good introductions for some of the existing methods.
Cons
- The novelty is limited. The basic idea of the proposed method is to simply concatenate the embeddings of two nodes (via activation separately) from both side of edges, which is straightforward and produces only marginal improvement over existing methods (the comparison of Figure 1 and Figure 3 would suggest this fact). The optimization algorithm is not novel either.
- Lack of detailed description and analysis for the proposed model S. In Section 5.2, only brief descriptions are given for the proposed approach.
- The selected baseline methods are too weak as competitors, some important relevant methods are also missing in the comparisons. For the graph embedding learning task, one of the state-of-the-art approach is conducting Graph Convolutional Networks (GCNs), and GCNs seem to be able to tackle this problem as well. Moreover, the target task of this paper is mathematically identical to the rating prediction problem (if we treat the weight matrix of the graph as the rating matrix, and consider the nodes as users, for example), which can be loved by a classic collaborative filtering solution such as matrix factorization. The authors probably need to survey and compared against the proposed approach.

---

### Official Review · AnonReviewer1 · 2017-11-26
**link weight prediction with pre-trained node embeddings**

**Rating:** 3
**Confidence:** 5

**Review:**

The authors propose to use pretrained node embeddings in a deep learning model for link weight prediction in graphs.
The embedding of the source node and the destination node are concatenated and fed into a fully connected neural network which produces the link weight as its output.
Existing work by Hou and Holder 2017 trains the same architecture, but the node embeddings are learned together with the weights of the neural network. In my professional opinion, the idea of using pretrained node embeddings and training only the neural network is not enough of a contribution.

Since the proposed method does not build on the SBM or pWSBM the detailed equations on page 2 are not necessary. Also, Figure 1, 2, and 3 are not necessary. Fully connected neural networks are widely used and can be explained briefly without drawing the architecture.

Pros:
+ interesting problem
+ future work. evaluation of embeddings is indeed a hard problem worth solving.

Cons:
- not novel

---

### Official Review · AnonReviewer2 · 2017-11-29
**Novelty is not enough**

**Rating:** 4
**Confidence:** 3

**Review:**

The paper presents a generic approach to graph link weight prediction problems based on node enbeddings. After introducing several existing methods, the paper proposes a "generic" link weight prediction approach that uses the node embedding produced by any node embedding techniques. Six datasets are used for evaluation.

Overall, the difference to the existing method [1] is minor. I don't think there is much novelty in the "generic" approach. More essential abstraction and comprehensive analysis is needed for a strong ICLR paper.

[1] Yuchen Hou and Lawrence B Holder. Deep learning approach to link weight prediction. In Neural
Networks (IJCNN), 2017 International Joint Conference on, pp. 1855–1862. IEEE, 2017.

---

### Decision · Program_Chairs · 2018-01-29
**ICLR 2018 Conference Acceptance Decision**

**Decision:**

Reject

**Comment:**

This paper does not meet the acceptance bar this year, and thus I must recommend it for rejection.